# Autism Spectrum Disorder and Remote Learning: Parents' Perspectives on Their Child's Learning at Home

Stephanie Hernandez * and Lisa D. Bendixen

Educational Psychology, Leadership, and Higher Education, University of Nevada Las Vegas, Las Vegas, NV 89154, USA; lisa.bendixen@unlv.edu
* Correspondence: hernas8@unlv.nevada.edu

**Abstract:** The sudden change in educational setting during the COVID-19 pandemic naturally raised questions about students' educational attainment. Access to in person teaching and special education services became restrictive and parents had to consider becoming more involved in their child's academic attainment. This exploratory case study examined parents' experiences in providing support to their child with autism spectrum disorder (ASD) during quarantine where they engaged in remote learning. The parents were interviewed mid 2021 using Zoom. Interview questions were generated through the lens of Vygotsky's Zone of Proximal Development and Scaffolding. The participants consisted of parents (n = 3) with a child who was medically diagnosed with autism. The children (n = 3) were in primary school and between the ages of 5–9. Three major themes emerged in all of the families. Families had a successful transition experience where they were initially concerned with remote learning but eventually became accustomed to the new learning setting. Secondly, families expressed having a productive home learning environment using a routine that incorporated a strong team of educational and therapeutic professionals helping in different areas of their child's development. Thirdly, all parents provided ongoing parental assistance that established guidance throughout remote learning.

**Keywords:** remote learning; autism spectrum disorder; COVID-19; parental involvement





## 1. Introduction

The global COVID-19 pandemic led to the universal restricted access of school grounds and resulted in students transitioning to a remote learning environment. Students with autism spectrum disorder (ASD) are typically eligible for special educational services that provide additional support during school. The delivery of multiple resources became limited, and parents had to consider how their child's academics and services would be incorporated in their new educational environment [1].

*Study Objectives and Research Questions*

The feelings and thoughts of parents were explored in relation to their children's educational attainment. This investigation sought to identify if parents helped with maintaining the supports that aid their child's cognitive growth during remote learning. Therefore, the following research questions were addressed:

1. What is the experience of parents who are responsible for remote learning with their child who has autism spectrum disorder during a global pandemic?
2. Do parents scaffold their child's learning at home? If so, how?
3. What instructional strategies do parents use to assist their child's learning?
4. What was the nature of the child's learning over the course of remote learning?

The preliminary findings can contribute to the literature in parental involvement by providing insights to develop practical guides that inform parents of ways to guide their child's cognitive development at home. Specifically, this research may be of interest to

parents who homeschool their children with autism spectrum disorder [2]. There is a need for more literature for parents looking to transition their child to homeschooling. Students with autism often go through stress in a new environment with unfamiliar expectations [3,4]. The current study shows students with autism transitioning to a new educational environment that included a communicative interdisciplinary team, strong parental involvement, and a curated educational environment that aligned to their needs. These aspects, that were incorporated into the participants transition with the help of their parents, may be helpful for families who seek homeschooling. The findings may also be helpful for parents whose children are not receiving special services.

## 2. Positionality Statement

For the past five years, the researcher has worked with children and teens who are diagnosed with autism spectrum disorder as a Registered Behavior Technician (RBT). A part of their work included clear communication with parents on their child's progress and difficulties. Once the COVID-19 pandemic impacted the educational system, they noticed a difference in both parental involvement and topics of discussion during sessions. The researcher considered that parents may have some obstacles, such as time constraints, in helping their child learn, or have limited educational supplies, or have trouble creating a focused learning environment at home. They also considered that many parents with a child who has autism are exposed to varied resources that they can refer to and use to help with teaching at home. Overall, the researcher was open to explore the experiences each parent had because they acknowledged that the parents' background and their child's characteristics may also play a role. Therefore, a qualitative case study was used to explore different alternatives without a set position in mind.

## 3. Literature Review

### 3.1. Parental Involvement and Scaffolding in a Traditional Education Setting

Before the wide-scale implementation of remote learning, parental involvement in elementary school children's education typically consisted of helping with homework, school projects, or a daily reading log. Parent involvement has an important impact in students' skills development when parents become interested in using evidence-based guidance gained from other service providers with which they communicate [5]. A meta-analysis of 14 studies that manipulated the training of parental involvement in homework found higher homework completion, fewer homework problems, and possible improvement in academic performance in elementary school children [6]. Parental involvement and academic success had a positive association for elementary school age children especially in verbal achievement outcomes. The parental involvement strategy of setting rules also had the highest association in achievement-related outcomes [6]. Students can benefit from parental training in homework by the parents being present as an additional educational support system for the child. Students are more accepting of parental involvement when parents are knowledgeable and properly deliver support that is suited to their needs [7].

A useful concept when providing assistance to a child comes from Vygotsky's (1978) theory of Zone of Proximal Development (ZPD). It explains that social interactions with a skillful adult are essential for a child's cognitive growth. Scaffolding is a key concept of Vygotsky's ZPD that refers to a more experienced peer or adult actively guiding a child's cognitive growth within their zone of proximal development. The difference between scaffolding and other forms of guidance is the gradual fading and elimination of support until independence [8]. Scaffolding is deemed ineffective for promoting cognitive growth when it does not mentally engage and elicit the creation of new information. The information must be in the child's area of knowledge, while also ahead of it, so that it may lead to development in the targeted area [9]. Therefore, the person who scaffolds a child's learning must be aware of the child's ZPD boundaries. Modeling, hints, cueing questions, open questions, and providing a piece of the solution are ways to scaffold student learn-

ing [10]. Non-scaffolded instruction would be asking students to work on activities such as meaningless rote calculations that do not engage their deep thinking [11].

The previous literature shows that parental scaffolding can aid child development in many ways [12]. Parents can help improve their child's academic and cognitive abilities as well as their executive functioning and verbal acquisition [12]. Compared to educators, it was expected that parents would use scaffolding more casually and occasionally. Parents can partake in scaffolding during daily routines, teaching moments, and working on homework. During remote learning, ideally children use video conferencing on their electronic devices to communicate with their educators and classmates, while the parent scaffolds their learning. Parents who use Scaffolders must recognize their child's understanding and current area of cognitive development to effectively promote learning [8].

### 3.2. Effective Scaffolding in a Traditional Education Setting Using Technology

Students using online remote learning may experience confusion, which can be mitigated by scaffolds, and, in turn, develop associated cognitive abilities. In the classroom, instructor and peer scaffolding are especially useful when a student is hesitant or unsure of technology-displayed scaffolds [8]. The software used in technology-enhanced classrooms can be difficult for some students to handle. Younger children may need more assistance when approaching technology-enhanced environments due to their interpretation of presented information such as instructions for an online task. On the other hand, some students, particularly young students, may be motivated to learn when educational materials are delivered through an electronic device [8]. Individuals who scaffold learning may also have trouble effectively implementing scaffolds. Teachers have encountered trouble with promoting student-centered learning, as well as limited resources and a lack of time, while using technology in the classroom [10]. Hence, barriers in technology-enhanced learning may hinder the providing of scaffolds to students.

Students benefit from scaffolds that consider their thinking processes. For example, students gain knowledge from guidance that considers prior experience and tailors problems to contexts that suit their interests and needs [10]. The necessary scaffolding strategies during problem solving activities in everyday classrooms, that utilize technology-enhanced learning, include problem identification and engagement (vivid descriptions and questions related to students), problem exploration (helping to navigate evidence), presentation and communication (guiding thinking, considering and evaluating other evidence and solutions) [10].

It is also important for students to have adequate skills so technology-enhanced scaffolds may be useful in grasping educational content [13]. Students may need time to learn to click and navigate through a page before attempting to learn the content that is presented [14]. Thus, certain technology-enhanced scaffolds may not be appropriate for a child's ZPD. Teaching while students use electronic devices is not widely common in all classrooms, therefore, implementing technology-enhanced learning is a process that may come with some obstacles. Parents who are not familiar with technology-enhanced learning may have difficulty while scaffolding their child's learning in the online environment.

### 3.3. Parental Scaffolding at Home Education

Students' remote learning environment is accompanied by online instruction that parents must then implement at home. Transitioning to remote learning has brought more responsibility to parents in educating their children at home. In the home, parents can take the extra step to adjust the environment appropriately to ensure it aids their child's cognitive growth. Hurlbutt [2] interviewed parents who homeschool their children with ASD. In most cases, the mothers were the primary caregivers and planners, the fathers were active in carrying out tasks such as completing paperwork for tax purposes, planning activities, and teaching lessons. Parents took the opportunity to accommodate the home environment for sensory difficulty, flexible scheduling, positive socialization, and incorporation of the child's interests in schoolwork and studies [2].

Andrews and Wang [15] found that a mother's scaffolding played a role in her child's emergent science competencies. The mother scaffolded her child's learning at home by providing resources, asking questions, selecting activities, and explaining the issues at hand. Parents can provide effective scaffolds in the home by verbally delivering them to their child [15]. The interaction between parent and child requires the child's active involvement as well [15]. The way a student processes parent scaffolds and whether they interpreted them correctly is also important in guiding them towards their goal.

Neumann, Hood, and Neumann [16] conducted a case study of a mother scaffolding her child's literacy ability prior to entering school. The authors provided descriptions of the child's literacy development from age 2–6. The mother scaffolded her child's writing and letter knowledge at home. Writing skills were scaffolded by using environmental print and household objects to learn letter names, sounds, and shapes. The mother would use directional language as the child would copy environmental print. Accessibility to environmental print in the home creates opportunities for literacy learning [16]. The authors emphasized that parents could help their child move along the zone of proximal development with appropriate scaffolding. Scaffolding in the home could also elicit more interactions with parents and eliminate the need for parents to purchase or prepare special educational materials. Children's familiarity with everyday environmental print (e.g., food labels and letter print in toys) also makes it easier to learn. The authors found that the child continued their use of directional language, print motivation, and private speech while learning, with or without the mother's assistance [16].

The mother in the study had a background knowledge in education, which may have helped her effectively navigate her child's learning [16]. However, many parents do not have the advantage of a background in education and may not know how to effectively use scaffolds in the home. Additionally, not all parents may be aware of academic learning opportunities available in the home. There is also an association between a higher level of education and using scaffolds that are more adaptive to the individual student stage of development [12]. Parent education can be an additional support for some families with children with autism and other disabilities [17]. Parents that have a higher level of education can also have greater problem solving skills [12].

Yet, many parents may not have the advantage of a higher education, thus, recognizing that most parents may not have the appropriate tools to educate their child at home presents a relevant topic to be addressed in the homeschooling literature.

*3.4. Parent Assisting Students with Autism in Traditional vs. Remote Learning*

Autism spectrum disorder (ASD) is a neurodevelopmental disorder with a range of characteristics including social and communication difficulties and restrictive and repetitive behaviors. In school, some students who have ASD depend on a routine to help them transition throughout the day. Knowing when and where the next task is happening can reduce anxiety and create structure [18]. Often, change is not welcomed by some students who have ASD. Due to the global pandemic, the learning environment that students were accustomed to was suddenly disrupted. In school, some students with ASD also like quiet places in and out of the classroom that give them a sense of security and help them manage their tasks throughout the school day [18]. Individual tailor-made support, low student/staff ratios, and a welcoming school environment are also viewed positively by parents of students with ASD [18]. Classroom teachers are aware of students' needs and necessary instructional modifications [19]. Additional support can include supplementary classroom materials, more time on tasks, designated seating area, and adjustments to help avoid distractions [20].

Parents who are interested in homeschooling their child with autism may deal with obstacles that will make it difficult to implement academics at home [3]. Parents have found it difficult to implement rules and authority during homeschooling [3]. Maintaining homeschooling schedules and procedures can be difficult to follow initially. Another transitional challenge would be behavior management, where the child might display

behaviors that are unwanted by the parent [3]. These challenges can then bring on stress and concerns to parents [3]. Students with autism can experience anxiety from transitions because their routine behaviors have been disrupted [4]. Transitioning students to a new environment also requires them to develop new behaviors [4]. Thus, preparing for a change in educational environment can reduce their anxiety. Students with autism benefit from learning environments that are predictable [4]. If students can understand and anticipate what is coming they will have an easier time transitioning to a new learning environment [4]. Implementing a structure during the transition that includes individualized planning, family involvement, organized curriculum, student development, and an interdisciplinary team can be beneficial for students with autism [4].

Not only can the sudden change in home environment lead to stress but there can be additional stressors for families with children with a disability because of their dependency and limitations [17]. Responsibility increases for parents and more of their attention is given to unwanted behaviors at home and less for leisure time and recreational activities [17]. Students diagnosed with autism often experience communication difficulties or delays which can make it harder to express themselves to others. Students with autism may have trouble communicating with their parent and the parent may have trouble understanding their child's requests [21]. Communication between parent and child must be strong and clear so that the child may continue to progress through learning assignments. Children with ASD often work with special services providers on developing their communications skills [21]. Furthermore, it has been shown that a parent's perception of the amount of social support can have a greater effect on coping with their stress than the severity of their child's difficulties [17].

During remote learning, parents can attempt to meet their child's educational needs at home that include necessary accommodations that their child's education-based service providers and teachers implement at school. When effective transition planning is in place, students with autism benefit from specific detailed information about their new settings [18]. Parents can create a learning environment that is similar to the classroom routine with which the students are familiar. These familiarities may include a designated desk or work area, a laptop, computer, iPad or any other electronic device to work on remotely.

As parents work on remote learning with their child, they will be using an any available electronic device to communicate with school staff. Parents' perspective of scaffolding their child's handling of iPad touchscreen tablets was reflected in the experience of their child, showing that, if their child had positive behaviors and feelings, so did the parent [22]. Parents have also shown appropriate scaffolding that aligns with their children's developmental gains. The main supports used by parents were verbal, physical, emotional verbal, and emotional physical [22].

## 4. Materials and Methods

A qualitative exploratory case study and collected interview data were used to explore parents' involvement in their child's educational development [23]. Participants were initially recruited through purposeful sampling from a local business that employs Board-Certified Behavior Analysts and Registered Behavior Technicians (RBTs) that directly provide Applied Behavior Analysis (ABA) therapy to children and young adults who are diagnosed with ASD. A participant recruitment flier was given to three different gatekeepers, managers at their respective local business. The gatekeepers distributed the flyers to potential participants using their preferred distribution method.

Collected data included an audio recording from each interview, interview transcripts, anecdotal notes from interviews, schoolwork documents, progress reports, Individualized Education Programs (IEP), and follow-up question responses.

### 4.1. Participants

This case study included three parents with children who are diagnosed with autism spectrum disorder and who moved to remote learning during school closures due to the

pandemic. All parents were physically at home during their child's remote learning. All three parents and their children had no previous experience with remote learning at home. Each parent completed a participant information sheet that requested some demographic information about themselves, their child, and their family (see Table 1). Olivia, age 41, reported that her child, Arnold, age 5, who attended kindergarten, had achieved some academic goals before remote learning. Arnold's household size consisted of 4 individuals. Arnold's remote learning was also supported by his father and grandmother. Emily, age 39, reported that Philip, age 8, who attended 2nd grade, achieved some academic goals prior to remote learning. Philip's household size was 3 individuals. Emily reported that she was the sole responsible party for Philips remote learning at home. Lisa, age 37, reported that her child, Noah, age 8, who attended 3rd grade, had multiple special educational supports. Noah had achieved some academic goals before remote learning. Lisa also indicated that his grades were B average. Noah's household size was 2 individuals and Lisa was the main responsible party for his remote learning. Lisa also had experience with homeschooling her child before remote learning. All three children were in elementary school and received online classes that were provided by their school teacher. All the children had support from special education service providers, including OT, PT, and SLP, and had ABA services as well. All primary caregivers were mothers and the children's main support at home. All their parents were at home during the day while their child had remote learning. Olivia and Lisa both worked from home and Emily was a stay-at-home parent. All parents stated they were middle class families (see Table 1). All children were diagnosed with ASD in early childhood.

**Table 1.** Participant demographics.

| Participant Name | Gender | Ethnicity | Age | Occupation | Education | Household # | Child |
|---|---|---|---|---|---|---|---|
| Olivia | Female | Hispanic | 41 | Licensing Manager | 16 y | 4 | Arnold, Age 5, Kindergarten |
| Emily | Female | Caucasian | 39 | Stay-at-Home Mom | 12 y | 3 | Philip, Age 8, 2nd Grade |
| Lisa | Female | Caucasian/ Hispanic | 37 | Wellness Advocate | 16+ y | 2 | Noah, Age 9, 3rd Grade |

*4.2. Interviews*

Each semi-structured interview was conducted via Zoom call. The tree and branch interview structure was followed by dividing the research problem into equal parts and covering it with interview questions [24]. Interview questions were developed to answer the main research questions of the study (see Figure 1). Once the first main research question was fully answered with corresponding interview questions, then the next main research question continued. Each participant was audio-recorded while follow-up questions and probing was administered throughout the interview [24]. Anecdotal notes were taken as well. Additional follow-up questions were carried out through email or by telephone depending on the participants preferred method of communication.

*4.3. Personal Documents*

Personal documents included completed paper packets, written assignments, as well as progress reports. All participants shared completed school work examples and a form of their child's progress report or a recent IEP. An Individualized Education Program (IEP) is an agreement of eligible special education instruction and services that is documented between a teacher, faculty, specialist, parent, and, when appropriate, the child. The progress report and IEPs reflected the children's learning that the parents noted in interviews. Schoolwork constituted tangible documents completed during remote learning that schools provided for students. Documents such as homework, schoolwork, and any other permanent products that were used for remote learning added contextual data and verified or disconfirmed interview data [25] Anecdotal notes and personal documents went

through the same analytic process that interview transcripts went through. To analyze data, the process of open, axial, and selective coding was used to develop concepts and discover themes [26].

☐ **Research Questions**

☐ **Interview Questions**

**1. What is the experience of parents who are responsible for remote learning with their child who has autism spectrum disorder during a global pandemic?**

- o What do you think about the overall home learning structure?
- o How do you feel before, during, and after working on schoolwork with your child?

**2. Do parents scaffold their child's learning at home? If so, how?**

- o What kind of interactions do you and your child have while working on remote learning?
- o How often are you with your child during remote learning and for how long?
- o What do you usually do while your child is engaging in remote learning? What is your role?

**3. What instructional strategies do parents use to assist their child's learning?**

- o Do you prepare for your child's online class, if so, how?
- o What is your child's learning setting like at home?
- o How long does your child engage in learning at home?
- o Do you provide any resources for your child during class?

**4. What was the nature of the child's learning over the course of remote learning?**

- o How do you think your child is responding to schoolwork at home?
- o Have you noticed anything different that your child does since starting remote learning?
- o Have you noticed any changes in their learning?

**Figure 1.** Research questions and corresponding interview questions.

*4.4. Data Analysis*

Transcription of interview audio recordings was carried out using Temi. Interview transcripts, anecdotal notes, and personal documents were analyzed using ATLAS.ti statistical software [25]. An inductive approach was used when coding where meaning was produced from the information gathered [27]. The organization of codes, code groups, and themes followed a pattern of hierarchy [27].

Data Analysis Coding and Levels of Analysis

Levels of analyses were revisited as open codes, axial codes, and selective codes, which were peer reviewed and reanalyzed as new concepts emerged. Codes that were meaningful were reused and codes with the same name or meaning were merged [28]. Categories were built bottom-up as codes started off as being very specific and then moved onto a more general concept and onto a theme that encompassed all specific meanings of quotes [28]. Pictures were coded with the same codes to either a section or sections of a picture or the whole picture [28].

Open coding was used to attach labels or names that described a concept related to the quote segment it was attached to. Research questions were referenced while searching for quotes to code. For example, I attached code (sitting next to) "I'm sitting next to him and I'm assisting him" (Olivia) and "I sit next to him and I, help him" (Emily). Open coding resulted in an initial list of 38 codes.

Axial coding was used to create categories by identifying connections between open codes. For example, routine was grouped with setting, prepare, regular basis, and learning structure. I would group codes and put a label on them that represented the concept for

which they stood. For example, codes such as praise, introducing fun, for the better, and rewards were labeled as the group code positive reinforcement.

Axial codes were then used to create selective codes which represented main themes. For example, positive reinforcement, prompting, and proximity were grouped together under the major theme of ongoing parental assistance. Parents used prompting as a technique while helping their children with remote learning. Proximity was also an important concept because it was part of the parental assistance. Grouping similar concepts led to the emergence of themes that reflected the main research questions. Emerging themes were analyzed and irrelevant subthemes were deleted, thus resulting in three strong and supported themes (see Table 2).

**Table 2.** Major themes summary.

| Themes | Category | Codes |
|---|---|---|
| Successful Transition Experience | Adaptation | emotions, versus classroom, surprised, solid at first, child change in feeling, socialize, child learning, used to it, uncertainty, struggle at beginning |
| | Resources | experienced homeschooler, home structure, additional helper |
| | Team Approach | need of routine, communication, team effort, services, experts modeling assistance |
| Productive Home Learning | Routine | used to it, ASD, learning structure, for the better, setting, regular basis, prepare, for child best, sitting next to, team approach |
| | Learning Progress | adapted, change, child change in feeling, negative child behavior, positive child behavior, positive child feeling, school better, positive child behavior, child learning, child behavior, before pandemic, versus classroom, remote for the better, used to it, ASD |
| Ongoing Parental Assistance | Proximity | assisting, scaffold, communication, learning structure, sitting next to |
| | Prompting | ASD, sitting next to, experts modeling assistance, scaffold, assisting |
| | Positive Reinforcement | for the better, introducing fun, praise, rewards |

*4.5. Procedures to Address Trustworthiness and Credibility*

Member checking and peer debriefing by the research team ensured validity of collected data by having participants read, review, clarify, and elaborate on the information they provided [29]. Triangulation was also used to confirm that information gathered was consistent and true [26]. All data were analyzed using ATLAS.ti for an effective and secure analysis [25]. All data were archived in a secure drive to which only the researchers had access.

**5. Results**

*5.1. Theme 1: Successful Transition Experience*

The first theme that emerged was a successful transition experience. A team approach and the provision of resources were keys to creating a successful remote learning environment at home. Parents shared their experiences during the transition to remote learning. Emily stated: "In the beginning, I was extremely worried. You know, I had tons of emotions. Like how am I going to do this?" Olivia was concerned about incorporating remote learning at home: "So, it was really hard for me because I also work and my husband works . . . So it was hard . . . You have to be with him all, all the time". Olivia also mentioned: "Well it was really hard at the beginning because he won't stay still".

Parents and their children had to adjust to communicating and interacting with teachers and other service providers in a different format. "The OT and the speech sessions would end with both of us in tears. He was just having a hard time adjusting of course, because he's autistic and he wants to go to school" (Lisa). Emily mentioned that: "Like

sometimes I think he hears the questions or the requests, he just daydreams off. I don't know. I wish I knew what he was thinking or, you know". Emily talked about her child using the iPad to communicate with others: "He mainly only uses it, um, with other people or if I seriously can't figure out what he's trying to ask me". Olivia also mentioned that the youngest child, Arnold, had trouble maintaining focus on the computer screen: "And so we're always Arnold sit up straight, Arnold pay attention, Arnold look at the teacher hahaha. But I think it's part of the distance education thing". Parents expressed that their child had some communication difficulties, like communicating with others through the computer screen, or, at times, that they wondered what their child was thinking. Overall, parents knew how to help their child communicate with others and with each other.

Parents described multiple resources that helped them with the transition. Teachers and therapists performed well in organizing the online learning environment so that the parent could easily access materials. Lisa mentioned that: "it's all done through his school, so everything is already uploaded . . . and you just like click". In addition, all parents put effort into preparing materials and creating a designated area for remote learning. Olivia said: "Yeah like whatever material they sent him from school, or the crayons in a box under his desk or the pencils and we usually have everything handy around his desk".

Parents also expressed how others' involvement helped with the transition to remote learning. Parents mentioned that multiple professionals would assist with their child's learning and development. Special education service providers and teachers, including parents, created the academic learning environment at home. All three children had an online learning class schedule that their teacher set up to follow at home. Their parents then implemented that schedule into their home routine and set up an area where the child carried out their studies. All three children received ABA therapy, such as many other students with autism who receive some type of additional therapy that supports a variety of daily living skills [30]. Lisa noted that "he receives services for reading, writing, and math. And then on top of that, he also gets the OT, the speech, the physical therapy." Emily talked about her child's RBT: "she's here for all of his school. So in between his school, in between those 20-min periods or whatever, he's doing ABA". Emily mentioned that Philip's teacher was well prepared and collaborated with his BCBA, including attending IEP meetings. Lisa also mentioned that Noah's previous teacher suggested ABA therapy and other services:

> So we've just been lucky because they were the ones who actually told me about ABA and like about sports, social, or, you know, things like that. So I wouldn't have known any of that without them.

Lisa seemed to have a more positive transition period compared to the comments of the other two parents: "The teachers had everything in place. All the Google meets links. I mean, everything was perfect. There was no issues." Lisa had experience with homeschooling her child before the switch to remote learning. Lisa used her homeschooling experience as a resource to help with the transition and really enjoyed the whole experience.

All the parents expressed that they had rough times at first because it was a big change for them and their child to have them spend the day working on academics and therapy at home instead of at school. Their special education support system and teachers helped the children with their education and parents felt supported too. Parents expressed they were supplied with the support that their child needed to make it feel like an environment that they could manage.

*5.2. Theme 2: Productive Home Learning Environment*

The second theme that emerged was creating a productive learning environment at home. Parents shared techniques, tools, and resources that helped with learning in the daily happenings of remote learning. The implementation of a routine created a solid structure for the children to continue their studies. Parents and children primarily followed the teachers online remote learning schedule at home. Parents understood that their child

needed structure and thrives when there is consistency among all individuals. Parents also shared their thoughts and feelings on the setting they prefer for their child. Olivia said:

> Thank God. The teacher figured a way that he teaches, like for 20 min and then they break and then another 20 min and then break and, and so on. So it was pretty cool because that way we can also do our things and he has a rest.

Parents noticed how their child followed and engaged in a routine when presented with certain stimuli. For example, Lisa mentioned that "he knows that Schoology is, is where we do our, the homework and the schoolwork". Emily also felt this learning setting was helpful for her child: "I feel like when Philip comes into this room, he knows it's time to work or it's time for school ... "

Parents created a productive learning environment where they witnessed their child's learning progress. For instance, parents prepared learning/therapy materials ahead of time for a smoother transition during remote learning including a quiet room, light music, snacks, and printed out materials. In this changed setting, parents witnessed their children's ability to adapt. Olivia mentioned that her child began to accept more requests than prior to remote learning, where the child was reluctant to accept parental involvement. For example, Olivia stated that Arnold began to communicate a lot more with his parents, including a greater understanding of their requests.

> Um, before he didn't want to accept any, any new stuff or learning that came from dad and mom ... And then when he, when we switched to distance education, and that I can be here more at home now. I'm a little bit more involved and he accepts learning from mom and dad.

Parents took notice of their child's behavior and emotions in the classroom and in the remote learning setting and described the benefits of each. Parents discussed their preference between going to school and remote learning. Lisa preferred that her child go to school: "I will say that he does, in my opinion, he does better at school because yes, like I'm strict with him, but I'm always still gonna be mom." Olivia stated that her child: "likes school, but right now, he does like it, but, he's not the same. I think he's more enthusiastic when he goes to school".

Emily was able to witness her child's learning progress as well: "So with him being home and me being next to him and helping him learn these things, I know where he is academically." Emily also said: "I know in the classroom from what his teacher said, that there would be a lot of protesting and things during the day in school, but we don't have that here." and "It's kind of like him having his own one-on-one aid versus being in a classroom where they might have up to 10 kids and one teacher and one aide". Overall, parents preferred the environment that resulted in the best learning for their child.

Parents expressed that their environment had a manageable routine with and in which their child could engage. Their teachers, IEP team, and strong parental involvement made it an effective learning environment and parents expressed that their child was carrying on with the educational curriculum they would be learning at school. The parents mentioned that the behaviors and feelings of their child were positive during learning activities at home. They also expressed that they prefer aspects of a remote learning environment.

*5.3. Theme 3: Ongoing Parental Assistance*

In the current study, there was a lot of ongoing parental assistance. Parents used proximity, prompting, and positive reinforcement to manipulate the environment to best teach their child. The proximity between parent and child was a key factor in delivering important assistance. Emily had the opportunity to stay close by her child:

> I don't know if I mentioned I actually stay home with him. So I'm very lucky with that. Also I know there are a lot of parents that work and may not be able to do as much as I'm able to do. So, I think that has helped a lot too.

All parents stayed close in proximity by sitting next to their child as they helped with communicating with teachers online. It was necessary for Olivia to sit next to Arnold, the youngest child in the study: "if I'm not there, he wouldn't respond to anything and I have to also be muting and unmuting the computer, stuff like that." The teacher was not there to directly structure the learning environment; thus, parents structured it and were more hands-on with implementing academic material.

Parents also followed prompting techniques that their therapists and teachers would use such as verbal, physical, and visual prompts. Parents did not show scaffolding but instead used guidance they labeled as prompting. Scaffolding would be guidance from the parent that is intended to be in the child's ZPD and leads to independence. All parents mentioned helping the child complete a task with their help but not leading them to independence in a skill. Prompts helped students process incoming information. For example, Emily mentioned a visual prompt that the RBT would use: "they use token systems, he understands all that." Lisa described using a visual prompt herself: "every parent should have a little whiteboard because it's, so life-changing when you're doing math with the kids, especially with kids that have special needs." Lisa also described her use of schoolwork prompting: "so we do it until it's correct. Um, because right now at this stage he won't be doing it independently." Olivia described how she would deliver verbal prompts to direct her son's behavior and how she gained confidence in herself to be able to teach him over time. Prompting was not reduced to lead to independence. This is to be expected, because identifying the appropriate level of assistance for their child's zone of proximal development was not a primary concern. Parents did not show scaffolding but instead used prompting in this study.

Parents also carried out actions that elicited more positive behavior such as scheduling certain activities first or giving a break. Lisa provided an example: "I made it fun because they did have a lot of schoolwork. So we would go outside and go swimming until he had his class, . . . and he did great with that." In addition, parents delivered praise so the child felt supported, positive, motivated, and engaged while working on schoolwork, so that it could be completed and they would not become overwhelmed. "Philip is the type of kid that he doesn't like to be wrong. So, he really feeds well into praise, so he wants that, 'Hey, good job', or, uh, tickles or anything" (Emily).

Parents also presented rewards for their children so they would have something to look forward to as they completed their work. "I have, like different color whiteboard markers so he can choose the different color that he wants when we're doing either math or writing an essay, which is fun to keep him still engaged" (Lisa). Parents chose to introduce a fun environment, so their child was engaged while learning and was not overwhelmed. Lisa described how she tried to read aloud to make reading more fun for her son:

> So I generally try to read quite a bit of it because if he's on the screen for too much, we're going to run into meltdowns. And so we kind of avert that by me reading and I do like fun voices and all this kind of stuff to help keep him engaged in the story.

Parental involvement by being nearby providing direct guidance and positive reinforcements was helpful for their child in completing assignments and they expressed it led to positive encounters for them and the child. Parents supplied prompts and one-on-one assistance directly to their child while engaging in remote learning.

Overall, parents had become accustomed to the remote learning environment and made positive remarks about this learning context. Parents in this study described a successful transition experience that initially started with some struggles but developed into a productive home learning environment with the help of multiple resources and their ongoing assistance.

All parents expressed feelings of nervousness or stress during certain aspects of transitioning to remote learning. All three parents expressed that their children displayed negative behaviors during the transition. However, all parents also expressed that the teachers and service providers helped them transition with their cooperation. All their

teachers had prepared an online class schedule and their service providers had prepared online therapy sessions. Each parent had prepared school supplies, assignments, and schedules to follow at home. Their behavioral therapy sessions had also helped with the children with varied skills at home. Every one of the parents were very involved and were prepared with an open mind to guide their child's learning in a way that was best suited to their needs and wants.

## 6. Discussion

This study explored the experiences parents had while their child transitioned to remote learning as a result of school closures due to the COVID-19 global pandemic. Parents shared their thoughts, feelings, and perspectives from their child transitioning to remote learning at home. Three common themes were found in the current study. The first theme was a successful transition experience that showed parents taking on added responsibility and providing support supplied by educational professionals to create a new educational environment. The second theme was creating a productive learning environment at home. Lastly, the third theme was ongoing parental assistance. Families experienced these three major commonalities throughout remote learning. Parents were set up for success where they felt equipped to guide their children's academic learning at home. Previous studies have also indicated that some parents struggled more than others while others embraced more responsibility [31]. Some parents preferred virtual instruction, however, parents of students with higher intense needs struggled more [32]. In some cases, parent and child had emotional stress due to the transition to remote learning and parents noticed negative behaviors and emotions in their children [33]. Some negative aspects that parents have mentioned included change in roles, increased responsibility of family, and psychological stress from uncertainty [31]. Parents' emotional state was also impacted by tiredness and mental disposition [33]. Some families also faced health problems [31]. In the present study, parents had moments where they experienced stress while working on assignments due to communication but, overall, parents knew how to effectively communicate with their child during schoolwork [34]. Many families have shown resilience and innovation from the sudden change in educational environment, which can be beneficial [32]. Resilience and overcoming change is helpful for students' development that is, to an extent, dependent on their individual circumstances [35].

### 6.1. Home Learning Routine

All three families in the current study had a productive home learning environment. Most students had online classes for more than 2 h a day during the pandemic [36]. Parents created a designated space accompanied with necessary supplies during specified times for their children to work on remote learning activities and therapy sessions. Parents manipulated the educational environment at home with the goal of assisting the student's learning. Parents had the opportunity to incorporate their child's needs during remote learning [2]. All parents considered their child's educational and functional needs. Parents set up a specific area that represented learning time and removed distractions, plus included tools that maintained their focus. Students with ASD generally prefer to learn in environments that are not overwhelming [18], and parents in the current study were able to accommodate. Parents ensured that their child was in a quiet area of the house while other individuals were home. Students with ASD also prefer a welcoming environment at school, which parents re-created at home by providing positive reinforcement during remote learning [18]. The three children thrived with a routine as it allowed them to be mentally prepared to work on subsequent tasks, which helped them transition smoothly between tasks. Previous research has emphasized the importance of routine during remote learning [35]. A routine provides an environment without unexpected events that may lead to discomfort [18].

### 6.2. Team Approach

Parents had the support of special educational professionals and service providers to create a routine for their children. Educators took a step by step approach in helping parents initially adjust to remote learning [37]. Firstly, educators tried to make contact with the parents then slowly introduced simple interactions with child and parent [37]. Then, they incorporated some online activities with no harsh expectations of deadlines [37]. Teachers gave room for leniency so parents and children would not be overwhelmed [37]. The third step was bringing in academics [37]. Special educators and school-based specialists adjusted their practices to best fit remote learning. Educators and specialists would adjust service minutes, add individualized contingency learning plans to IEP, and eliminate social goals at times [32]. It has been suggested that parents stay in close contact with special education services during remote learning [33]. Teachers have helped by providing individual and group meetings with parents via Zoom [37]. Teachers have also used Zoom meetings to supply parents with advice and tools to support their child through the individual obstacles they were facing [37].

### 6.3. Parental Guidance during Remote Learning

The third theme that emerged was ongoing parental assistance, where parents positioned themselves right next to their child to guide while working on schoolwork. Parental involvement increased because, in most cases, it was unavoidable that parents support their child for at least 30 min a day [36]. New research on parents' outlook on distance learning with children with ASD found that parents liked having more quality time with their family [31]. Parents can monitor their children's emotions and behaviors while working on remote learning and therapy and take notice of what elicited positive behaviors and what did not. All parents in this study had the more favorable circumstance of staying at home when their child had remote learning and used that time to increase educational support. Online learning can have positive outcomes for some children with adequate support at home [35]. One participant in the current study, Lisa, had a background in education, which may have helped with implementing effective learning techniques. A parent who has a background in education has knowledge in delivering academic material in a way their child can effectively learn [2,12,16]. The parents helped with any confusing aspects of technology use by navigating the computer screen and accessing online materials as well as setting up accessibility to the teacher [8]. In one case, the youngest child was often redirected to listen or look at the teacher on the screen and was not always engaged. He may still be adapting to remote learning or learning how to dedicate time to learning in general. Parents of younger children are often more involved than older students [36]. Older children receive less scaffolding when using touchscreen tablets compared to younger children [22,38]. Parents provided sufficient support to their children in this study, however, teachers have witnessed inequity among student support and the level of parental involvement among families was variable [37].

### 6.4. Parents' Use of Prompts

In this study, parents discussed how they provided ongoing assistance in their child's academics and varied behaviors. This study focused on the social interactions between the more knowledgeable person (i.e., parent) guiding the less knowledgeable individual (i.e., child) towards cognitive growth, specifically investigating how parents guide their child's learning in an area they are familiar with and can accomplish with their assistance. Ideally, parents would lead their child to independence in this area, thus increasing their cognitive growth. Scaffolding is a process that can take time and multiple steps and shifting of instruction to be more useful [12]. Parents used prompting as a main tool to assist and guide their child during remote learning. Parents did not show scaffolding but did show prompting, which can be a useful guidance tool as well. Parents did not show scaffolding because their guidance was not intended to guide the child to learn the skill so they can accomplish it on their own.

In the current study, other professionals involved in the child's education included their special education teacher, occupational therapist, physical therapist, and behavioral therapist. Therapy targets areas the child may be having difficulties or delays in, such as communication and social skills, repetitive behaviors, or restrictive interests. These professionals use different approaches when guiding learning and one of these strategies is prompting. Prompts come in many forms such as physical, verbal, visual, or modeling. Prompts are delivered at the beginning of a task, and the intensity of the prompt is modified according to the child's skill level [39]. Therapists maintain knowledge of a child's current development and can supply the appropriate prompting to assist at a level where the child can progress to the point of independently doing the skill. Using ABA therapy tools, such as prompting and increasing the time, with their child may have helped with them accepting parental involvement [21]. During therapy, it is important to explicitly state why certain prompts or assistance are implemented during a specific task so parents are aware and knowledgeable about their child's development. When parents are informed, they can help implement similar assistance at home to maintain a consistent environment.

Parents were aware that their child needed assistance in certain areas. This may be because parents were in close contact with their child's service providers, and specific areas and goals were indicated. It may also be that the proximity between parent and child introduced opportunities to learn about specific behaviors upon which their child was working. Parents also know specific quirks and characteristics that others may not see or be familiar with that may help with effectively communicating and delivering information to their child. However, parents were using prompting and would continue to prompt with the same intensity without decreasing the intensity. Parents' assistance was, at times, too intense or too unfamiliar and did not give the child the opportunity to develop the skill on their own independently. A longer study would observe parents utilizing scaffolding techniques while acknowledging the child's ZPD with the goal of cognitive growth and independence.

### 6.5. Theoretical and Practical Implications

Remote learning at home was an effective learning environment for students with ASD. With the appropriate resources and team support, parents were able to aid their child's learning at home. However, remote learning at home may not be effective as a long-term educational setting. Children with autism need opportunities to practice their social skills, especially if they are known to have difficulties in that area. Parents can use this study to reflect on the practical techniques used during remote learning. Professionals can use these parents' experience to develop guides that help parents who are homeschooling a child who has ASD. Parents who prefer to be more hands-on may be interested in pursuing an online hybrid charter school or specialty program for students with autism that includes additional parent training necessary for home learning. With a hybrid model, parents can have more control over their child's education and use their additional insight on their unique characteristics to aid learning.

Parents looking to transition into homeschooling may come across barriers in helping their child adjust to learning academics at home [3,4]. Parents can increase parental involvement by initiating communication with their child's support team. Parents can seek awareness of the tools that professionals are using, such as prompts, and attempt to apply them as well. It would be important for parents to lean on others to help carry some of the responsibility by making sure that their child has adequate assistance in all areas and that their support team is consistent.

### 6.6. Limitations of Present Study
#### 6.6.1. Sample Size and Characteristics

Some limitations of the study included the sample size and the lack of in-person interviews and direct observations of the phenomenon. A larger sample size may have supplied additional information from a variety of participants. This study included only

boys and the primary caregivers who supported their learning were all mothers. Therefore, there is some lack of representation of children from different ages, genders, and diversities of the ASD spectrum.

Parents in this study experienced a positive outcome from remote learning with their children who have ASD. The experiences may have differed for parents who have different demographics than the participants in this study. Parents with a higher education may have additional knowledge that would help in implementing a productive learning environment for their children, and a familiarity with specific resources to help their child's academics.

Parents in this study had access to additional resources that other parents and children may not have access to, potentially due to low socioeconomic status or lack of access to therapy services covered by health insurance. Additionally, due to lack of school funding in some areas, children may not have adequate educational services and resources relative to schools that have greater funding allocated to educational services for students with disabilities. Thus, a study that includes families with different educational and socio-economic backgrounds may have outcomes that differ from the participants in this study.

### 6.6.2. Additional Data Collection Methods

This study could have benefited from direct observations by collecting additional means of data that could not be shared through an interview. By not conducting in person interviews, the mannerisms or facial expressions that show emotion may be overlooked during parents' responses. Solely collecting interview data may give the researcher one subjective perspective and objective occurrences could be missed during this phenomenon. However, the global pandemic introduced these limitations when conducting human subject research.

Personal documents may be subjective and parents may have only shared items that show the extreme of a situation, not the typical case. Parents' comments and answers may have been subjective or influenced by uncontrollable variables [23]. Another personal document might have been more informative than the ones they decided to share.

A longitudinal study, that began earlier during the transition to remote learning, could have gained fresh perspectives and emotions on the topic. Conducting interviews as the participants continue remote learning and then transition to hybrid learning could also provide additional perspectives on this phenomenon.

### 6.7. Future Research Directions

Future research should consider more studies that include parents and children with ASD working in a remote learning environment. Future research should also look into identifying individual student skills in students with ASD and implementing parent scaffolding to aid the development, and possibly independence in relation to, said skills using a longitudinal study. It would be valuable to continue research in training parents to utilize scaffolds as a technique to teach their children different skills while also considering individual student characteristics. Possible future research could look into the relation between parents using prompting techniques from a therapeutic setting and scaffolding from an educational setting, specifically using them towards skill independence. Future research should consider a larger sample to investigate the different teaching approaches and techniques parents use during remote and hybrid learning. More research should also focus on students with learning disabilities or impairments that may have obstacles that inhibit their learning progress while engaging in technology-enhanced environments. Additional research should also look into students' active role while accepting scaffolds, including students diagnosed with ASD.

**Author Contributions:** Conceptualization, S.H.; methodology, S.H.; software, S.H.; validation, S.H. and L.D.B.; formal analysis, S.H.; investigation, S.H.; resources, S.H. and L.D.B.; data curation, S.H.; writing—original draft preparation, S.H.; writing—review and editing, S.H. and L.D.B.; visualization, S.H.; supervision, L.D.B.; project administration, S.H. and L.D.B. All authors have read and agreed to the published version of the manuscript.

**Funding:** This research received no external funding.

**Institutional Review Board Statement:** The research was approved by the UNLV Social/Behavioral IRB. Project Title: [1705715-3] Autism Spectrum Disorder and Remote Learning: A Parents Perspective on Their Child's Learning at Home.

**Informed Consent Statement:** Informed consent was obtained from all subjects involved in the study.

**Data Availability Statement:** The data are not publicly available due to privacy and ethical restrictions.

**Conflicts of Interest:** The authors declare no conflict of interest.

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
