# Peer review of "Autism Spectrum Disorder and Remote Learning: Parents’ Perspectives on Their Child’s Learning at Home"

_education, doi:10.3390/educsci13070716_

Round 1

Reviewer 1 Report

This is a qualitative study evaluating 3 parents’ involvement in their autistic children’s educational development within remote instructional contexts during the COVID-19 pandemic. Participating families were purposely selected from a local business of Certified Behavior Analysts and Registered Behavior Technicians (RBTs). Families were interviewed and interviews were audio recorded from each interview.

The authors clearly present their methods for analysis of the interview data. The methods are carried out in a systematic manner. However, my primary concern is in regard to the topic of the study. Specifically, I’m wondering whether and how the findings from this study are meaningful given that children are no longer being educated remotely due to the pandemic. Rather, inquiry centered on what families learned and carried out during the pandemic and how they have continued to use these skills /strategies with their children may be a more meaningful direction of study. Overall, it is unclear how this study informs or contributes to the larger research base.

Although the themes that emerged from the data are interesting, they are very narrow in range -- primarily ABA-based. This is understandable given that the sample of participating families were recruited from an ABA establishment. However, this means that the sample is quite biased since all families were involved in ABA-based interventions with their autistic children. Is it possible that during the pandemic the families implemented strategies they had previously learned or were currently learning through the ABA services they were receiving? Hence, the theoretical and practical implications fall outside the range of this study, as findings do not generalize beyond the context of this study.

Furthermore, the literature review centers on scaffolding yet scaffolding is not reflected within the emerging themes. Better alignment between the literature review and the results/discussion section is needed, as the literature review does not provide a clear foundation for the study’s findings.

There also seems to be missing citations throughout the literature review to support the authors’ claims/ ideas and terminology is sometimes conflated – such as cognitive development vs academic growth/development. These are different areas of development. Paragraph structure and organization within the literature review is also difficult to follow at times.

Readability of the results section is difficult at times. It is recommended that the authors first summarize the findings and then provide specific quotes/examples to further support the findings. For example, how do the examples provided relate to larger theme, “Successful Transition” (Theme #1).

Providing more detail about the participating children in the study would be helpful for interpreting the study findings and further understanding the varying home contexts of the 3 children. Focusing on a specific area of learning rather than all academic learning, or providing detail on the academic/school learning context would also be helpful. It is unclear how similar or different these children’s and families experiences were – this makes grasping the themes difficult.

Reviewer 2 Report

Dear Authors

The authors are to be commended on an excellent and well written topic, very relevant to the current shift to home-schooling particularly for autistic students. I would suggest the authors consider including some literature on transitions which can be very problematic for autistic students if not planned and managed with supports. As this is a large focus of the shift from school to home this would make the article more evidence based.

I would suggest a thorough proofe read especially in the literature review as there are many incomplete sentences see for example lines 73 and 91.
